

# The impact of sleep loss on sustained and transient attention: an EEG study

Lucienne Shenfield[1], Vanessa Beanland[2], Ashleigh Filtness[3,4] and
Deborah Apthorp[5,6]

[1] Research School of Psychology, Australian National University, Canberra, ACT, Australia
[2] Department of Psychology, University of Otago, Dunedin, New Zealand
[3] School of Design and Creative Arts, Loughborough University, Loughborough, Leicestershire,
United Kingdom
[4] Queensland University of Technology (QUT), Centre for Accident Research and Road Safety –
Queensland (CARRS-Q), Kelvin Grove, QLD, Australia
[5] School of Psychology, University of New England, Armidale, NSW, Australia
[6] Research School of Computer Science, Australian National University, Canberra, ACT, Australia

Corresponding authors
Lucienne Shenfield,
lucienne.shenfield@gmail.com
Deborah Apthorp,
dapthorp@une.edu.au

## ABSTRACT

Sleep is one of our most important physiological functions that maintains physical
and mental health. Two studies examined whether discrete areas of attention are
equally affected by sleep loss. This was achieved using a repeated-measures within-
subjects design, with two contrasting conditions: normal sleep and partial sleep
restriction of 5-h. Study 1 compared performance on a sustained attention task
(Psychomotor Vigilance task; PVT) with performance on a transient attention task
(Attentional Blink; AB). PVT performance, but not performance on the AB task,
was impaired after sleep restriction. Study 2 sought to determine the neural
underpinnings of the phenomenon, using electroencephalogram (EEG) frequency
analysis, which measured activity during the brief eyes-closed resting state before the
tasks. AB performance was unaffected by sleep restriction, despite clearly observable
changes in brain activity. EEG results showed a significant reduction in resting
state alpha oscillations that was most prominent centrally in the right hemisphere.
Changes in individual alpha and delta power were also found to be related to changes
in subjective sleepiness and PVT performance. Results likely reflect different levels of
impairment in specific forms of attention following sleep loss.

Attentional blink, PVT, Alpha EEG, Delta EEG

# INTRODUCTION

## Sleep and attention

Healthy sleep habits are imperative to maintaining physical and psychological health and
wellbeing (*American Academy of Sleep Medicine, 2014*; *Hirshkowitz et al., 2015*), while
sleep loss is associated with physical and mental health problems (*American Psychiatric
Association, 2013*). Costs associated with sleep loss in Australia were estimated at
$36.4 billion in 2010 (*Deloitte Access Economics, 2011*). Despite this, recent evidence
suggests that the number of hours people habitually sleep per night has continued to
decline (*Knutson et al., 2010*; *Matricciani, Olds & Williams, 2011*).

**How to cite this article** Shenfield L, Beanland V, Filtness A, Apthorp D. 2020. The impact of sleep loss on sustained and transient attention:
an EEG study. *PeerJ* 8:e8960 DOI 10.7717/peerj.8960

As discussed by *Owens (2009)*, considerable empirical evidence indicates that the central nervous system centres which regulate sleep overlap with those that regulate attention/arousal. This suggests that a disruption in one system is likely to be reflected by a disruption in the other system. Research suggests that sleep loss tends to impair attention, but 'attention' encapsulates a wide range of different paradigms and phenomena. As noted by *Whitney & Hinson (2010)* and *Raz (2004)*, many studies have broadly claimed that sleep deprivation negatively impacts all areas of attention. However, few empirical studies have sought to evaluate each separate interrelated process within the individual cognitive tasks. Even simple cognitive tasks typically involve multiple underlying cognitive processes, and some task components may be more susceptible to the effects of sleep deprivation than others.

## Vigilance and the psychomotor vigilance task

Vigilance is defined as the ability to sustain attention over an extended period of time (*Oken, Salinsky & Elsas, 2006*). Sustained, focused, endogenous, and vigilant attention can be viewed as synonymous terms (*Carrasco, 2011*). Vigilant attention has often been examined in the context of sleep loss and it appears to be highly susceptible to its effects (*Lim & Dinges, 2008*).

Vigilance tasks include the Continuous Stimulus Detection or Discrimination tasks (*Langner & Eickhoff, 2013*), Continuous Performance Test (*Riccio et al., 2002*), Go-No-Go test (*Ayalon et al., 2009*) and Psychomotor Vigilance Task (PVT; *Dinges & Powell, 1985*). The PVT involves a 10-min trial in which participants are instructed to respond to a stimulus (e.g. a millisecond counter) by pressing a button as quickly as possible when it appears at random intervals, usually 2–10 s. This randomised inter-trial-interval is integral in differentiating the PVT from a standard reaction time task, as it eliminates the opportunity for anticipation, forcing participants to be constantly vigilant (*Basner & Dinges, 2011*). The PVT (*Dinges & Powell, 1985*) has an extensive history of use in sleep-related research (*Lim & Dinges, 2010*), reliably finding that sleep loss results in poorer performance.

## Transient attention and the attentional blink

The concept of sustained attention is relatively well understood, but less is known about transient attention (*Liu, Pestilli & Carrasco, 2005*), which involves the ability to attend to brief, fleeting tasks or stimuli. Transient attention tasks are thought to facilitate greater attentional intensity than sustained attention tasks (*Carrasco, 2011*). Transient attention rises and decays quickly and typically has a duration of less than 10 s (*Suri et al., 2014*), with a peak intensity of 100–120 ms post initiation (*Carrasco, 2011*). Arguably, the most widely used test of transient attention is the attentional blink (AB) paradigm (*Dux & Marois, 2009*).

The AB occurs when a person is able to perceive a first target (T1) within a set of distractors, but has a reduced ability to perceive a second target (T2) when it is presented within 800 ms of T1 (see Fig. 1). The phenomenon was first documented by *Broadbent & Broadbent (1987)* and was named 'attentional blink' in 1992 by Raymond and

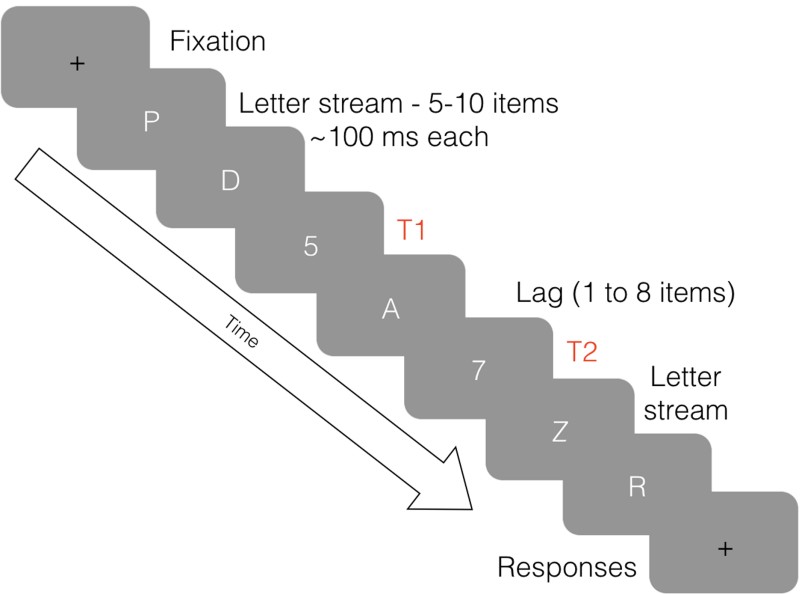

**Figure 1 An illustration of the attentional blink.** Eight frames of an RSVP stream in an AB task, depicting a lag of two between T1 and T2.

colleagues. This term does not refer to a physical blink, but rather, a momentary lapse in attentional ability (*Shapiro, Raymond & Arnell, 1997*). The event is typically measured using a Rapid Serial Visual Presentation (RSVP) task developed by *Potter & Levy (1969)*. During the task, distractor and target stimuli are displayed in quick succession, usually about 100 ms apart, to test information processing limits. For a thorough review of research and theories pertaining to AB, see *Dux & Marois (2009)*. While PVT performance has been extensively researched in relation to sleep, less is known about the relationship between AB performance and sleep loss.

## Electroencephalogram

Attentional blink research has frequently used Electroencephalography (EEG). EEG is a reliable and valid way of measuring the brain activity associated with different levels of consciousness (*Antonenko et al., 2010*), and has greatly advanced our understanding of the neural processes associated with sleep and wakefulness (*Strijkstra et al., 2003*). EEG is sensitive to activity of cortico-thalamic networks that underlie the dimensions of sleep and wakefulness (*Steriade, 1999*). For this reason, EEG has an extensive history of use in sleep and attention research (*Berka et al., 2007*; *Hoedlmoser et al., 2011*; *Strijkstra et al., 2003*). However, the majority of previous research using EEG to investigate the effects of sleep loss has focused on total sleep deprivation, rather than sleep restriction (*Borbély et al., 1981*; *Kaida et al., 2006*). Sleep deprivation is defined as a night in which no sleep occurs, while sleep loss/restriction is defined as a night in which a person sleeps less than is typical (*Reynolds & Banks, 2010*). In general, the effects of sleep deprivation are thought to be more severe than the effects of sleep restriction (*Reynolds & Banks, 2010*).

The measurement of neural activity using EEG enables the detection of synchronous activity in neurons which fire rhythmically in the cortex to produce distinct EEG
waveforms (*Tong & Thakor, 2009*). Neural oscillations vary in their amplitude and frequency, and they are generally separated into five bandwidths, referred to as delta (0–4 Hz), theta (4–8 Hz), alpha (8–12 Hz), beta (12–40 Hz) and gamma (40–100 Hz). Sleep has been linked to various wavelengths including alpha, theta and delta (*Benca et al., 1999*; *Boonstra, Daffertshofer & Beek, 2005*; *Borbély et al., 1981*; *Cajochen, Foy & Dijk, 1999*; *Harmony et al., 1996*; *Kaida et al., 2006*; *Klimesch, 1999*; *Strijkstra et al., 2003*). Additional findings regarding neural activity have linked alpha oscillations to visual attention (*Capotosto et al., 2009*; *Ergenoglu et al., 2004*; *Mathewson et al., 2011*), temporal attention (*Hanslmayr et al., 2011*), and the AB (*Händel, Haarmeier & Jensen, 2011*; *Maclean & Arnell, 2011*; *MacLean, Arnell & Cote, 2012*; *Zauner et al., 2012*). Subjective sleepiness and PVT performance have been linked to delta and theta oscillations (*Hoedlmoser et al., 2011*).

## STUDY 1

Study 1 investigated the impact of sleep loss on sustained and transient attention, using the PVT and AB paradigms. Prior research has shown that the two tasks are equivalent in terms of task difficulty/engagement (*Shenfield, Beanland & Apthorp, 2020*). Consistent with the abovementioned literature, it was hypothesised that performance would be impaired on (i) the PVT and (ii) the AB after sleep restriction.

## STUDY 1 METHOD

### Participants

Twenty-four adults (12 female; $M_{age}$ = 26.0 years, SD = 3.9) were recruited from among the staff and students at Queensland University of Technology (QUT). Study inclusion criteria were that participants were aged 18–30 years and regularly slept 7–8 h/night. Study exclusion criteria were current smokers, sleep/medical/mental health problems other than minor illnesses, high caffeine users (>5 caffeinated drinks/day), regular daytime nappers, and shift-workers. All participants received AUD$60 on completion of the study. The study was conducted with full ethics approval from the QUT Human Research Ethics Committee (HREC; Protocol #1300000793).

### Design and procedure

The study used a repeated-measures design. Participants completed two testing sessions: (i) after a normal night's sleep (NS) and (ii) after sleep restriction (SR) to 5 h, with session order counterbalanced between participants. The sessions were scheduled approximately 1 week apart. To ensure the standardisation of the sleep restriction, participants reduced their sleep to 5 h by delaying bedtime by 3 h, so that all experienced early night sleep restriction, as opposed to late night sleep restriction, which can affect attention differently (*Zerouali, Jemel & Godbout, 2010*). To ensure compliance, participants kept a sleep diary for three nights prior to each testing session, estimating their sleep onset, morning wakening and rising times. They also phoned an answering service prior to going to bed and after waking each morning, and the times were used to

verify the diary entries (*Ibáñez, Silva & Cauli, 2018*) and confirm that the participants typically slept 7–8 h/night.

Participants were instructed not to consume alcohol from 18:00 h the night prior to the study and not to consume caffeine on the day of the study. They were instructed to have breakfast as usual and a light lunch. They were asked to arrive to the study at 14:00 h, as this has been identified as an optimal time to measure the effects of sleep loss (*Lenné, Triggs & Redman, 1997*). At this time, they provided informed written consent, completed the demographics questionnaire, then rated their subjective sleepiness. They were required to undertake two computer-based tasks: (i) the AB task and (ii) the PVT. After a brief explanation of the tasks, they completed 10 practice trials of the AB task and 5 min practice of the PVT. After both practice sessions, the AB task was completed and then the PVT, with a short break in between the tasks.

## Apparatus

Both tasks were presented on the same computer using a 19-inch 100 Hz CRT monitor.

## Stimuli
### Psychomotor vigilance task

Stimuli were presented over a 10-min period, and the length of the interval between the stimuli ranged randomly from 1 to 9 s. Participants were seated at the computer with their dominant hand on a computer mouse. They were instructed to respond immediately when a digital millisecond counter appeared on the screen.

### Attentional blink

A custom AB program was written in Presentation (Neurobehavioural Systems, Inc.). Each trial was comprised of 10 items: eight distractors and two targets. Items were presented using RSVP in Times New Roman 50-point font. Distractors were all uppercase letters of the alphabet except those that might be confused with numbers (I, O, Q and S). Targets were the digits 2–9 inclusive. Two targets were presented in each trial. T2 appeared either immediately after T1 (lag 1) or after 1–7 distractors (lags 2–8). After each trial, participants used the number pad on the keyboard to indicate the two targets that had been presented. There were 160 experimental trials, 20 for each lag. The task took approximately 12 min to complete.

### Subjective sleepiness

All participants provided ratings of subjective sleepiness at the start and the end of each testing session using the nine item Karolinska Sleepiness Scale (KSS; *Åkerstedt & Gillberg, 1990*; *Kaida et al., 2006*).

## Statistical analysis

All data were inspected for normality and homoscedasticity. Subjective sleepiness (KSS) and PVT were compared between the SR and NS conditions using paired *t*-tests. Dependent measures derived from the PVT were mean reciprocal response time (RRT; 1/reaction time), standard deviation of RRT, slowest 10% RRT, number of lapses

**Table 1 Experiment 1: sleep data.** Duration of sleep and subjective sleepiness by testing session.

| | Normal sleep | Sleep restriction | *t*-test (2-tailed) |
|---|---|---|---|
| Previous night's sleep (minutes) | 467 (50) | 297 (18) | |
| Previous night's sleep (hours) | 7.8 | 5.0 | |
| KSS on arrival | 3.0 (0.3) | 5.1 (0.4) | $t_{(23)} = 6.03$, $p < 0.001^*$ |
| KSS prior to leaving | 3.9 (0.4) | 5.9 (0.5) | $t_{(23)} = 4.36$, $p < 0.001^*$ |

Note:
* Significance at $\alpha = 0.05$.

**Table 2 Experiment 1: PVT data.** PVT results-means and standard error of the mean.

| | Normal sleep | Sleep restriction | *t*-test (2-tailed) |
|---|---|---|---|
| Reciprocal RT | 4.14 (0.11) | 3.71 (0.15) | $t_{(23)} = 3.37$, $p = 0.003^*$ |
| RRT SD | 0.84 (0.16) | 0.72 (0.07) | $t_{(23)} = 0.68$, $p = 0.287$ |
| Reciprocal slowest 10% | 2.99 (0.12) | 2.49 (0.16) | $t_{(23)} = 3.90$, $p = 0.001^*$ |
| Mean number of lapses (sqrt) | 0.26 (0.11) | 1.24 (0.32) | $t_{(23)} = -2.80$, $p = 0.010^*$ |
| Mean number of false starts (sqrt) | 1.24 (0.20) | 0.80 (0.22) | $t_{(23)} = 2.07$, $p = 0.050$ |

Note:
* Significance at $\alpha = 0.05$.

(RT > 500 ms), and number of false starts (RT ≤ 100 ms). False start reaction times were not included in the RRT calculations. Data for lapses and false starts were square rooted to normalise the distributions.

The AB dependent measures was the percentage of T2 items correctly identified in cases where T1 was correctly identified (T2|T1), which was analysed using a two-way 2 × 8 repeated measures analysis of variance (ANOVA) testing condition (2) and lag (8). Greenhouse-Geisser adjustments were made in the case that the assumption of sphericity was not met. All analyses were evaluated using an alpha level of 0.05.

# STUDY 1 RESULTS

## Sleep characteristics

The results presented in Table 1 show that the SR manipulation was effective in significantly decreasing the amount of actual sleep and increasing the level of subjective sleepiness reported by participants.

## Psychomotor vigilance task

Psychomotor vigilance task results are presented in Table 2. SR significantly slowed participants' reaction time overall and in the slowest 10% of responses. There was also a significant increase in the number of lapses. However, SD of the reciprocal reaction time was not significantly affected and there was a marginally significant increase in the number of false starts after SR.

## Attentional blink

Repeated-measures ANOVA for T2|T1 accuracy revealed a significant main effect for lag, $F(2.8, 64.8) = 32.29$, $p < 0.001$, $\eta_p^2 = 0.58$, but no significant interaction between condition

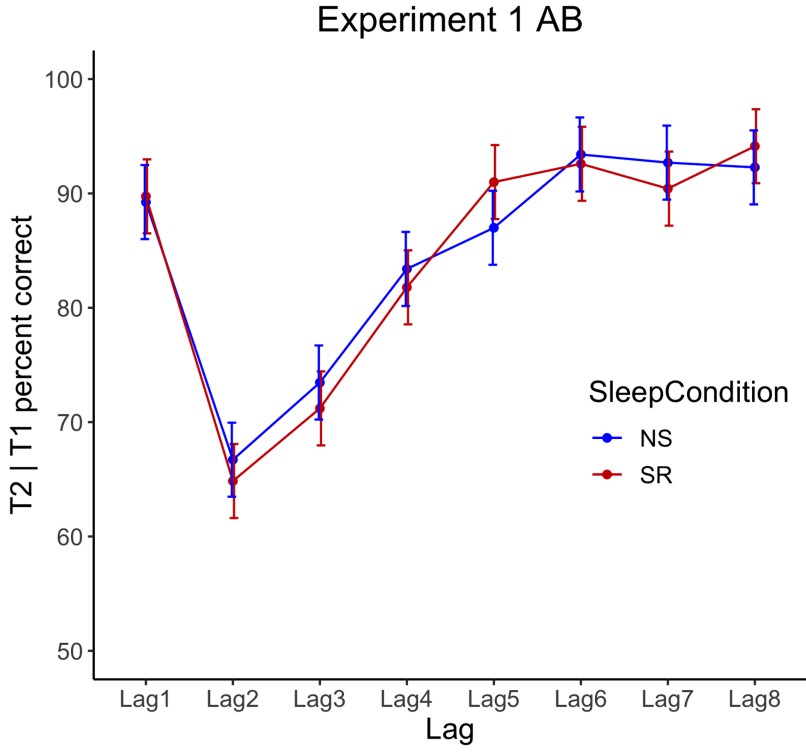

**Figure 2 Experiment 1 attentional blink performance.** Results show T2|T1 accuracy by lag for normal sleep (NS) and sleep restriction (SR) conditions. Error bars represent +/− 1 SE around the mean.

and lag, $F(4.7, 107.1) = 0.91$, $p = 0.497$, $\eta_p^2 = 0.04$, and no significant main effect of condition, $F(1, 23) = 0.06$, $p = 0.814$, $\eta_p^2 < 0.001$, see Fig. 2.

## STUDY 1 RESULTS AND DISCUSSION

The hypothesis that (i) performance on the PVT would be impaired during sleep restriction was supported, however the hypothesis that (ii) AB performance would be impaired during sleep restriction was not supported, as accuracy appeared to be robust to the impact of sleep restriction. This counterintuitive finding challenges the assumption that all aspects of attention are equally affected by sleep loss, which warrants further investigation. Consequently, a second study was designed to investigate the possible neural underpinnings of the observed differential decrement in performance, using EEG.

Specifically, Study 2 examined the neural underpinnings that are likely driving the observed attentional changes after partial sleep restriction. Based on the available literature, it was anticipated that sleepiness would correlate negatively with global alpha and positively with central frontal theta (*Strijkstra et al., 2003*), whereas delta and theta spectral power were expected to increase with sleep loss (*Hoedlmoser et al., 2011*). Thus, it was hypothesised that neural oscillations at rest will be related to the particular sleep condition and performance measures will replicate those observed in Study 1, that is (i) PVT performance will be impaired during sleep restriction, and (ii) AB performance will be unimpaired.

## STUDY 2 METHOD

### Participants

Twenty-four new participants (15 female; $M_{age}$ = 24.8 years, SD = 6.8) were recruited at the Australian National University (ANU) using Sona participant recruitment software, social media and word of mouth. The same study inclusion and exclusion criteria were employed as in Study 1. Additionally, individuals were excluded if EEG might be unadvisable (e.g. recent head injury, history of seizures, etc.). The same informed written consent procedures were used as in Study 1. All participants received AUD$30 upon completion. The study was conducted with full ethics approval from the ANU HREC (Protocol #2015/184).

### Design and procedure

A repeated-measures design was used. Participants completed two EEG testing sessions in counterbalanced order: (i) one after a normal night's sleep, and (ii) one after 5 h sleep; and the sessions were one week apart. Participants were asked to arrive at the testing session at 13:00 h, allowing 60 min to set up, so that they commenced the tasks at 14:00 h.

As a stronger test for compliance with the sleep instructions, participants were provided with a FitBit® Charge HR (*Diaz et al., 2015*; *Montgomery-Downs, Insana & Bond, 2012*) at least three days prior to the first EEG testing session, and they were instructed to wear it for the duration of the study. Additionally, they kept sleep diary estimates of sleep onset, morning wakening, and rising times. Although actigraphy is not as reliable as polysomnography at determining wake times (*Quante et al., 2018*), this provided an additional objective measure to supplement the sleep diary estimates and ensure participants complied with the instructions. As in Study 1, in the sleep restriction condition all participants were asked to delay their bedtime by 3 h, and to set their alarm for 5 h later. FitBit® data was assessed using Fitabase® software (*Diaz et al., 2015*). Data for the three nights of sleep prior to the testing days were used to confirm that the participants typically slept for 7–8 h.

At 14:00 h on both testing days, participants completed the KSS to assess their subjective sleepiness. All study tasks were conducted in a dark room to avoid peripheral distractions, and the researcher was absent from the room during testing. At the start of testing, they completed a brief exercise to ensure that eye movements were accurately detected, enabling their removal from further analysis. This involved moving eyes up, down, left, right and blinking on cue. Participants then completed the resting state exercise in which they were instructed to rest quietly for 2 min with their eyes open, and for 2 min with their eyes closed. After this, they completed the AB and PVT tasks. The order of the tasks was counterbalanced across the participants but it was consistent between the sessions. After completing the second EEG session, participants were verbally debriefed and given an opportunity to withdraw their data from the study, if they wished.

## Apparatus

### FitBit®

As an objective measure to supplement sleep diaries, participants were asked to wear FitBit® Charge HR activity monitoring devices for three days prior to each of the two EEG sessions.

### Electroencephalogram

A Compumedics NuAmps 40 channel EEG system was used to record electrophysiological activity. Electrodes were positioned according to the 10/20 system (*Jasper, 1958*), which has become the international standard in EEG research (*Jurcak, Tsuzuki & Dan, 2007*). The reference electrode was placed centrally at CZ. Mastoid electrodes were placed behind each ear, and eye movements (horizontal and vertical) were monitored by Electrooculogram electrodes. QuikGel® Electrolyte conductive solution was used to establish a connection between the epidermis and electrodes. To ensure a strong connection, impedance levels were set below 5 kΩ. Signals were amplified and recorded using the NuAmps digital amplifier at a sample rate of 1,000 Hz for offline analysis using Curry 7.0.9 by Compumedics Neuroscan, on Windows 7.

### Other equipment

Stimuli were presented on a 23.6-inch VIEWPixx liquid crystal display monitor with a refresh rate of 120 Hz and a resolution of 1,920 × 1,080 pixels. Behavioural responses for the PVT were collected using a RESPONSEPixx VPX-ACC-3100 five-button response box. Behavioural responses for the AB were collected using a Cedrus RB-830 response pad, with buttons labelled 2–9.

## Stimuli

Experimental stimuli were created using Psychophysics Toolbox version 3.0.12 (*Brainard, 1997*; *Pelli, 1997*) for MATLAB version R2012b.

### Psychomotor vigilance test

Psychomotor Vigilance Test presentations replicated the approach used in Study 1, but inter-stimulus interval length varied from 2 to 10 s.

### Attentional blink

Attentional blink task presentation replicated the approach used in Study 1, but fewer lags were assessed, to permit more repetitions of each lag. T2 was presented at a lag of 1, 3, 5 or 8 stimuli after T1. Each AB trial consisted of 18–22 items, including two targets. T1 was presented with jitter of ± 2 items, meaning that T1 appeared after 4–8 distractors. Items were presented in Helvetica font.

 Prior to the first test session, participants were given 12 practice trials at half speed, followed by 12 practice trials at full speed. During the second test session, they were given 12 practice trials at full speed. Auditory accuracy feedback was provided after each practice trial.

**Table 3 Experiment 2: ROIs.** Regions of interest and corresponding electrode groupings.

| Region of interest | Electrodes |
|---|---|
| Left frontal | F7, F3, FT7, FC3, FT9 |
| Left central | T3, C3, TP7, CP3 |
| Left occipital | T5, P3, O1, PO1 |
| Right frontal | F4, F8, FC4, FT8, FT10 |
| Right central | C4, T4, CP4, TP8 |
| Right occipital | P4, T6, O2, PO2 |

Following the practice blocks, 50 trials were included for each lag, resulting in a total of 200 trials per session. Rest breaks were offered after every 50 trials, with the duration decided by participants. The AB task took approximately 10 min to complete, excluding breaks.

### Subjective sleepiness

Subjective sleepiness was assessed using the KSS, this time administered once at 14:00 h in both sessions.

### EEG data analysis

Electroencephalogram analysis focused on the data gathered during the eyes-closed resting state conducted before the experimental tasks, hereto forth referred to as the resting-state. This decision was made to minimise artefacts in the data relating to eye movements. Resting state data were compared between sleep conditions. EEG data recorded during the eyes-closed resting state was useable in all cases. Analysis was achieved by isolating the desired epoch from the EEG data, which was then visually inspected for anomalous artefacts (e.g. jaw clenches, blinks, electrode pops, etc.), which were removed from further analysis. Data were baseline corrected, re-referenced offline to the common average reference, and filtered with a bandpass IIR filter between 0.5 and 47 Hz to remove drift and high-frequency artefacts. Only eyes-closed resting state data from 14 to 104 s were used, which eliminated muscle artefacts from closing the eyes. Electrodes were grouped into regions of interest (ROI) based on hemisphere (left and right) and scalp location (frontal, central and occipital; see Table 3). Alpha power was determined as the average power across frequencies between 8 and 12 Hz; delta power as 0.5–4 Hz; and theta power as 4–8 Hz.

Spectral power for the resting state was calculated for each channel using a sliding 2-s window with 0.75 overlap; each sample was windowed with a Hamming window to avoid edge effects, and power was extracted using a fast Fourier transform. Relative power was computed by dividing the relevant spectral band by the sum of all spectral bands for that individual (*Delorme & Makeig, 2004*; *Stoica & Moses, 1997*, *2005*). For each channel, the average power across all frequencies was calculated to produce an average amplitude spectrum. Spectral analysis was conducted using EEGLAB Version 13.4.4 (*Delorme & Makeig, 2004*) for MATLAB R2015a.

## Statistical analysis

All statistical analyses were conducted using SPSS Version 23. EEG data were analysed using repeated-measures ANOVA. The factors used in the ANOVA were the 6 ROI's and the sleep condition (NS or SR). ANOVAs were conducted for alpha, theta and delta frequencies. Means were compared (paired-samples $t$-tests and non-parametric tests) to determine whether sleep condition caused a significant difference in relative and absolute spectral power. FitBit®, sleep journal and KSS data were analysed using two-tailed paired-samples $t$-tests. In the event of skewed data, non-parametric tests were used.

Correlational analysis determined the most representative measure of attention performance in the PVT data, and paired-samples $t$-tests were used to determine if performance was affected by the sleep condition.

In assessing the AB data, non-parametric tests determined whether blink magnitude and performance for lags 1, 3, 5 and 8 differed significantly in the two conditions. AB magnitude was calculated by subtracting each participant's T2|T1 accuracy at lag 3 from their T2|T1 accuracy at lag 8 (*MacLean, Arnell & Cote, 2012*).

## STUDY 2 RESULTS

### Sleep characteristics

FitBit® data were inspected to ensure that participants: (i) habitually slept 7–9 h, (ii) achieved sufficient sleep in the NS condition (≥7 h), and (iii) genuinely restricted their sleep (≤5 h). Two cases were excluded on the basis of criterion (iii), and one was excluded due to criterion (i), leaving 21 valid cases. The PVT data were inspected for univariate and multivariate outliers using the procedures outlined by *Tabachnick & Fidell (2013)*.

Sleep diary and FitBit® data were visually inspected and tested for normality using the Shapiro–Wilk test. All variables met assumptions of normality, except SR sleep diary reports, $W(21) = 0.59$, $p < 0.001$, which is likely due to restriction of range (participants were instructed to sleep for exactly 5 h). Consequently, Wilcoxon Signed Rank Tests were used in all the analyses involving the SR diary reports. A Bonferroni-adjusted alpha level of 0.013 was used to reduce the likelihood of Type-I errors.

FitBit® data showed a significant difference between NS ($M = 8.2$ h, $SD = 0.9$), and SR ($M = 4.64$ h, $SD = 0.44$) conditions, $t(20) = 15.70$, $p < 0.001$; $d = 3.43$, 99% CI [2.01–4.89]. Similarly, there was a significant difference between the diary reports of sleep duration during NS (median = 8.5 h) and SR (median = 5 h) conditions, $Z = -2.43$, $p = 0.015$, $r = -0.53$. Results indicate that participants were genuinely restricting their sleep.

To test whether participants were accurately recording sleep, FitBit® data were compared with sleep diary estimates (see Fig. 3). No significant difference was found between the FitBit® data ($M = 8.2$, $SD = 0.9$) and diary reports ($M = 8.3$, $SD = 1.0$) in the NS condition, $t(20) = -0.52$, $p = 0.606$; $d = -0.11$, 99% CI [0–0.64]. However, there was a significant difference between FitBit® (median = 4.5) and diary (median = 5.0) data in the SR condition, $Z = -4.02$, $p < 0.001$, $r = -0.88$.

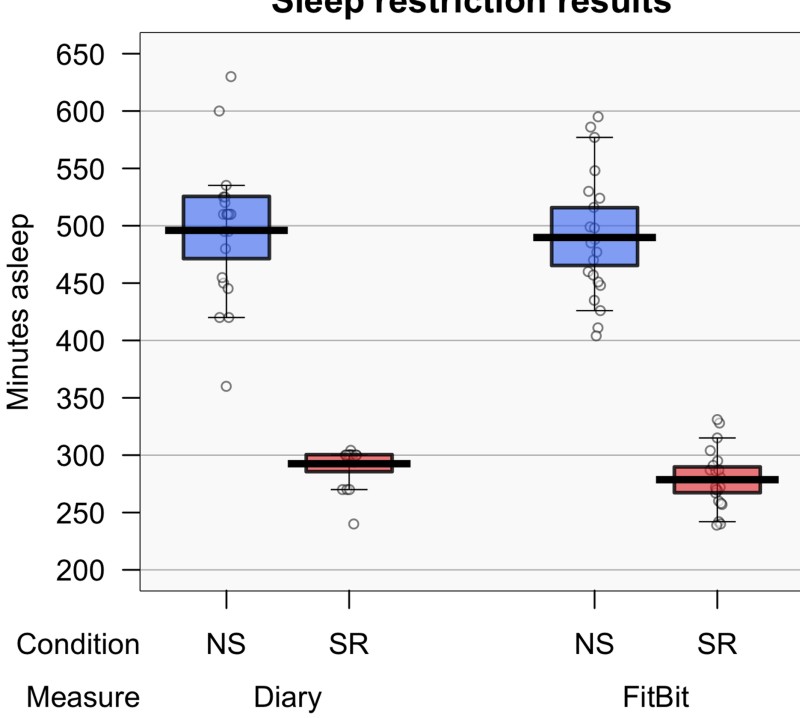

**Figure 3 Experiment 2: sleep data.** Recorded hours of sleep according to the FitBit® and sleep diary (left) and subjective sleepiness (right). Means are shown by black horizontal bars. Individual scores are represented by black circles, slightly jittered for clarity; coloured areas represent 95% Highest Density Intervals (HDIs), calculated using R's BEST (Bayesian Estimation Supersedes the *t*-Test) package, and vertical bars represent the 10th and 90th quantiles. 

## Subjective sleepiness

For KSS data, one multivariate outlier was detected using Mahalanobis distance (46.80, $p < 0.001$). This case did not report a change in sleepiness between sessions, indicating that the experimental manipulation had not affected their subjective sleepiness. KSS scores in the NS condition were normally distributed, but KSS scores in the SR condition violated assumptions of normality, $W(21) = 0.79$, $p < 0.001$. Visual inspection of the data determined that the identified multivariate outlier was driving the skew. Thus, the data were analysed with the outlier included and excluded. As the overall outcome was unaffected, the anomalous participant's data were retained. As visual inspection revealed that the distribution of the data was only marginally skewed, parametric and non-parametric tests were performed to compare the means. As the overall outcomes did not differ, only the results for the paired-samples *t*-test are reported here.

A paired-samples *t*-test determined that the KSS ratings differed significantly between NS ($M = 4.0$, SD = 1.8) and SR ($M = 6.9$, SD = 1.5) conditions; $t(20) = -7.83$, $p < 0.001$; $d = -1.71$, 95% CI [1.02–2.38]. Results suggest that the experimental condition significantly increased perceived sleepiness; see Fig. 4A.
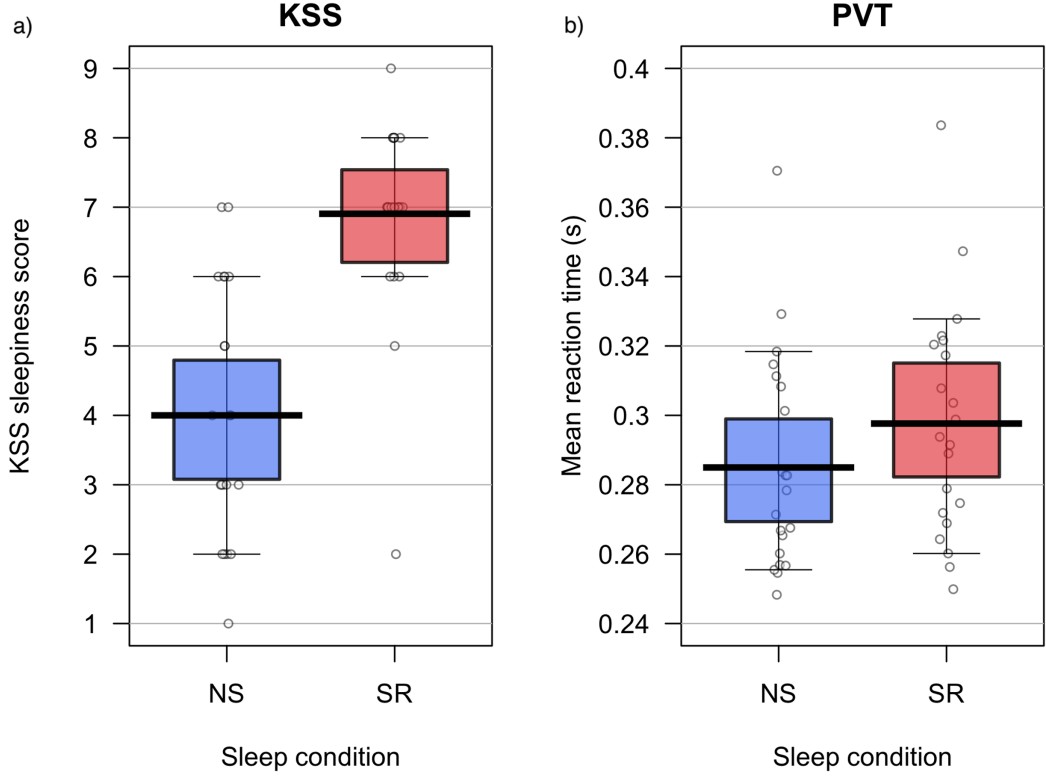

**Figure 4 Experiment 2: KSS data and PVT performance.** Karolinska Sleepiness Scale ratings (A) and PVT performance (B) during normal sleep (NS) and sleep restriction (SR). Means are shown by black horizontal bars. Individual scores are represented by black circles, slightly jittered for clarity; coloured areas represent 95% Highest Density Intervals (HDIs), calculated using R's BEST (Bayesian Estimation Supersedes the *t*-Test) package, and vertical bars represent the 10th and 90th quantiles.

## Main analyses
### Behavioural data (AB & PVT)

*Psychomotor vigilance test*

Some univariate outliers were detected in PVT performance, demonstrated by *z*-scores >|3.29| (*Tabachnick & Fidell, 2013*), but they were attributed to individual differences and did not affect the analysis, so they were retained. False starts, lapses, mean fastest 10% of RTs, mean slowest 10% of RTs, median RT, SD of RT, skew, mean RT, errors of omission, errors of commission and number of trials were recorded for each PVT session. Pearson product-moment correlation coefficients were computed to determine which measure of PVT performance was most representative of overall performance. Mean RT was most highly correlated with the other measures, indicating that it was the most representative measure of overall PVT performance. Mean RT was statistically and visually inspected for normality. Assumptions of normality were met. A paired-samples *t*-test determined that mean RT was significantly lower in the NS (*M* = 285 ms, SD = 0.03) than the SR (*M* = 298 ms, SD = 0.03) condition, *t*(20) = −2.49, *p* = 0.022, *d* = 0.54, 95% CI [0.09–0.99]; see Fig. 4B.

**Table 4 Experiment 2: AB lag comparisons.** Wilcoxon signed-rank test results for lags in NS and SR conditions.

| Lag | Sleep condition | Median | Z | Sig. | R |
|-----|-----------------|--------|------|-------|-------|
| 1 | NS | 0.93 | −0.29 | 0.775 | −0.06 |
|   | SR | 0.93 | | | |
| 3 | NS | 0.78 | −0.66 | 0.509 | −0.14 |
|   | SR | 0.81 | | | |
| 5 | NS | 0.89 | −0.36 | 0.722 | −0.09 |
|   | SR | 0.91 | | | |
| 8 | NS | 0.96 | −0.10 | 0.922 | −0.02 |
|   | SR | 0.95 | | | |

*Attentional blink*

Attentional Blink data were skewed; thus, non-parametric tests were conducted. Wilcoxon signed-rank tests, with a Bonferroni-adjusted alpha level of 0.013, showed that accuracy at each lag did not differ significantly in the NS and SR conditions. Wilcoxon signed-rank tests for NS and SR are displayed in Table 4. Mean proportion of correct responses per lag are displayed graphically in Fig. 4. Additionally, a Wilcoxon signed-rank test indicated no significant difference between the blink magnitude in the NS (median = 0.13) and SR (median = 0.10) conditions; $Z = -0.99$, $p = 0.322$, $r = -0.22$. The results are illustrated in Fig. 5.

### EEG data analysis

To reduce extreme skewness and kurtosis, absolute spectral power for alpha, delta, and theta bands were logarithmically transformed. Relative spectral power was normally distributed across all the frequency bands. One-way repeated measures ANOVA was used for all comparisons.

*Theta*

No significant difference was found between the sleep conditions in theta spectral power $F(1, 20) = 1.91$, $p = 0.183$; $\eta_p^2 = 0.06$, 95% CI [0.00–0.30] or relative power $F(1, 20) = 1.05$, $p = 0.317$; $\eta_p^2 = 0.05$, 95% CI [0.00–0.29].

*Alpha*

A significant effect of sleep condition was detected for alpha spectral power, $F(1, 20) = 8.18$, $p = 0.010$; $\eta_p^2 = 0.29$, 95% CI [0.02–0.53]. When ROIs were considered separately, the only difference to reach statistical significance (using a Bonferroni-adjusted alpha level of 0.008) was log of alpha in the central left and right hemisphere. Results indicate that there was a significant reduction in central spectral alpha across both hemispheres in the SR condition; see Table 5.

There was also a significant effect of sleep condition on relative alpha power, $F(1, 20) = 5.47$, $p = 0.030$; $\eta_p^2 = 0.22$, 95% CI [0.00–0.47]. When the results for ROIs were considered separately, the only difference to reach statistical significance was relative alpha in the

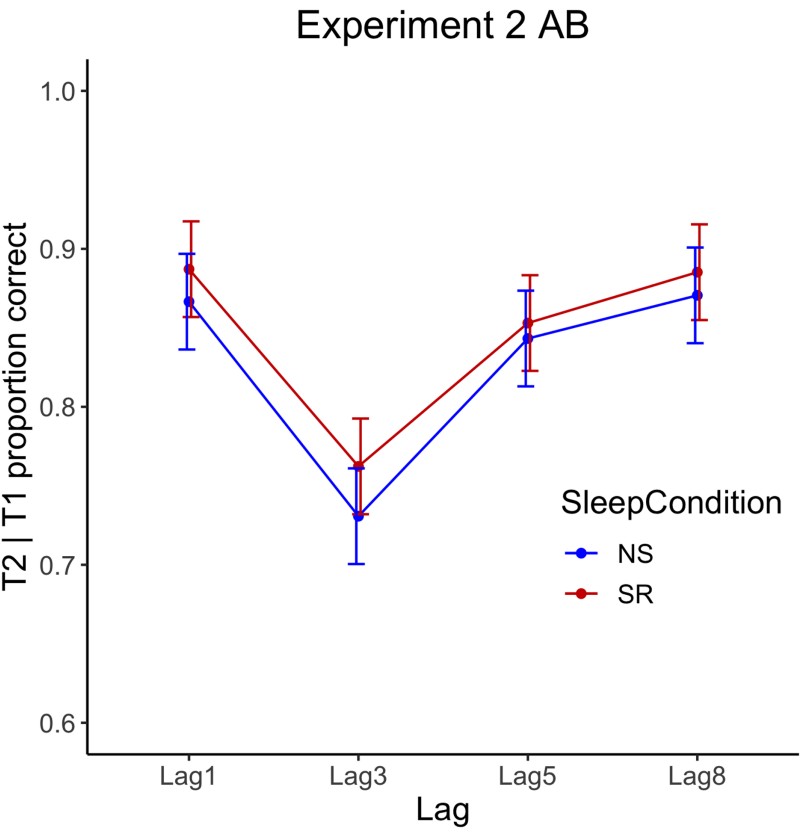

**Figure 5 Experiment 2 attentional blink performance.** Behavioural data for the AB task—the proportion of T2|T1 correctly identified by lag for normal sleep (NS) and sleep restriction (SR) conditions. Error bars show ± 1 SE around the mean.

central right hemisphere. Results indicate that there was a significant reduction in relative alpha spectral power in the central right hemisphere in the SR condition; see Table 5.

*Delta*

There was no significant effect of sleep condition on delta spectral power, $F(1, 20) = 0.53$, $p = 0.475$; $\eta_p^2 = 0.03$, 95% CI [0.00–0.25].

A marginally significant effect of sleep was detected for relative delta power, $F(1, 20) = 3.86$, $p = 0.063$; $\eta_p^2 = 0.16$, 95% CI [0.00–0.42]. When ROIs were considered separately, the only difference to reach statistical significance was relative delta spectral power in the central right hemisphere. Results indicate that a moderate increase of relative delta spectral power occurred in the central right hemisphere in the SR condition; see Table 6.

*Analysis of alpha and delta*

Figure 6 illustrates comparative changes in relative resting state frequency for alpha and delta bands, in normal sleep and sleep restriction conditions. A reduction in alpha and an increase in delta was observed during the sleep restriction condition. Figure 7 displays the regional locations of these changes.

**Table 5 Experiment 2: comparisons of absolute and relative alpha.** Results of paired-samples *t*-tests for log of absolute and relative spectral alpha.

| Region | Groups | | | | | | | | | |
|---|---|---|---|---|---|---|---|---|---|---|
| | Normal sleep (NS) | | Sleep restriction (SR) | | | | | | | |
| | *M* | SD | *M* | SD | *N* | df | *t* | Sig. | *d* | 99% CI |
| Absolute power | | | | | | | | | | |
| Log left frontal | 1.14 | 0.93 | 0.92 | 0.22 | 21 | 20 | 2.62 | 0.016 | 0.57 | [−0.06 to 1.19] |
| Log left central | 1.03 | 0.19 | 0.74 | 0.20 | 21 | 20 | 2.97 | 0.008* | 0.65 | [0.01–1.28] |
| Log left occipital | 2.12 | 0.24 | 1.88 | 0.26 | 21 | 20 | 1.91 | 0.070 | 0.42 | [−0.19 to 1.02] |
| Log right frontal | 1.07 | 0.19 | 0.86 | 0.20 | 21 | 20 | 2.23 | 0.037 | 0.49 | [−0.13 to 1.10] |
| Log right central | 1.20 | 0.21 | 0.86 | 0.22 | 21 | 20 | 3.12 | 0.005* | 0.68 | [0.03–1.32] |
| Log right occipital | 2.39 | 0.25 | 2.10 | 0.27 | 21 | 20 | 2.66 | 0.015 | 0.58 | [−0.05 to 1.20] |
| Relative power | | | | | | | | | | |
| Left frontal | 1.85 | 0.19 | 1.69 | 0.19 | 21 | 20 | 1.54 | 0.139 | 0.34 | [−0.26 to 0.93] |
| Left central | 2.21 | 0.19 | 1.96 | 0.16 | 21 | 20 | 2.48 | 0.022 | 0.54 | [−0.08 to 1.16] |
| Left occipital | 3.01 | 0.19 | 2.90 | 0.22 | 21 | 20 | 0.84 | 0.441 | 0.18 | [−0.40 to 0.76] |
| Right frontal | 1.74 | 0.17 | 1.60 | 0.18 | 21 | 20 | 1.54 | 0.140 | 0.34 | [−0.06 to 0.73] |
| Right central | 2.40 | 0.18 | 2.09 | 0.21 | 21 | 20 | 3.25 | 0.004* | 0.71 | [0.06–1.35] |
| Right occipital | 3.18 | 0.19 | 3.00 | 0.22 | 21 | 20 | 1.83 | 0.083 | 0.40 | [−0.21 to 1.00] |

Note:
* *p* < 0.008 indicates a significant difference. Confidence intervals were created using the Bonferroni corrected alpha level (0.008)

**Table 6 Experiment 2: comparisons of relative spectral delta.** Results of paired-samples *t*-tests for relative spectral delta.

| Region | Groups | | | | | | | | | |
|---|---|---|---|---|---|---|---|---|---|---|
| | Normal sleep (NS) | | Sleep restriction (SR) | | | | | | | |
| | *M* | SD | *M* | SD | *N* | df | *t* | Sig. | *d* | 99% CI |
| Left frontal | 2.19 | 0.16 | 1.32 | 0.17 | 21 | 20 | −1.13 | 0.273 | −0.25 | [0.00–0.82] |
| Left central | 1.96 | 0.16 | 1.87 | 0.13 | 21 | 20 | −1.54 | 0.140 | −0.34 | [0.00–0.92] |
| Left occipital | 1.20 | 0.14 | 1.25 | 0.15 | 21 | 20 | −0.53 | 0.602 | −0.12 | [0.00–0.68] |
| Right frontal | 2.27 | 0.17 | 2.44 | 0.16 | 21 | 20 | −1.46 | 0.161 | −0.32 | [0.00–0.90] |
| Right central | 1.53 | 0.12 | 1.80 | 0.15 | 21 | 20 | −3.28 | 0.004* | −0.72 | [0.01–1.36] |
| Right occipital | 1.09 | 0.13 | 1.20 | 0.14 | 21 | 20 | −1.47 | 0.158 | −0.32 | [0.00–0.90] |

Note:
* *p* < 0.008 indicates a significant difference. Confidence intervals were created using the Bonferroni corrected alpha level (0.008).

*Individual change in alpha and delta*

Individual changes in relative alpha and delta were analysed for the right central region, where the most significant changes in activity were observed, to determine if the individual changes in spectral power were related to the performance and questionnaire measures. Change scores were calculated as suggested by *Kirschfeld (2008)*. Change scores in relative alpha and delta were calculated for individuals as follows: alpha change = right central alpha SR–right central alpha NS; Delta change = right central delta

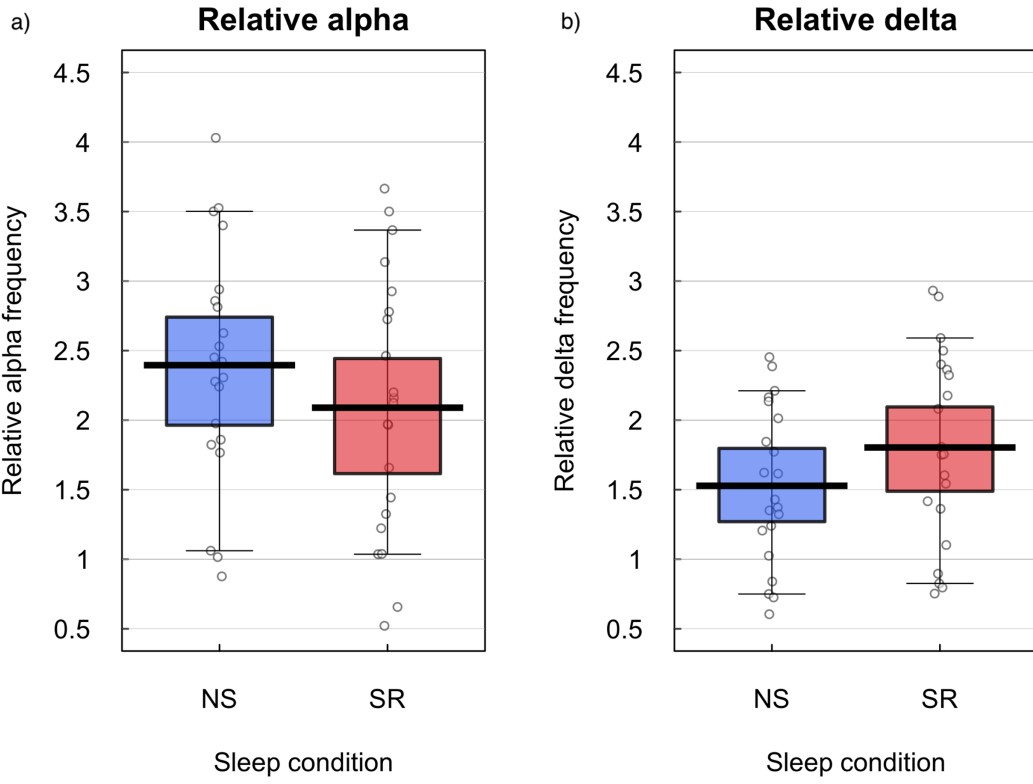

**Figure 6 Relative alpha and delta frequencies for normal sleep and sleep restriction conditions.**
Relative resting state frequency for alpha (A) and delta (B) bands in normal sleep (NS) and sleep
restriction (SR) conditions in the right central ROI. Means are shown by black horizontal bars. Individual
scores are represented by black circles, slightly jittered for clarity; coloured areas represent 95% Highest
Density Intervals (HDIs), calculated using R's BEST (Bayesian Estimation Supersedes the *t*-Test)
package, and vertical bars represent the 10th and 90th quantiles.

SR–right central delta NS. The same change calculation was used to calculate PVT change
(RT SR–RT NS), and KSS change (KSS SR–KSS NS).

Scatterplots in Fig. 8 display the correlations between changes in PVT performance and
changes in spectral power (left), and changes in subjective sleepiness and changes in
spectral power (right). Alpha change and RT change were moderately negatively
correlated, $r(20) = -0.49$, $p = 0.023$, suggesting that a greater reduction in individual
right central alpha during sleep restriction corresponded to a greater increase in RT in
individuals. Delta change and RT change were moderately positively correlated,
$r(20) = 0.42$, $p = 0.057$, suggesting that a greater increase in individual right central delta
during sleep restriction corresponded to a greater increase in RT in individuals. Alpha
change and KSS change were strongly negatively correlated, $r(20) = -0.51$, $p = 0.019$,
suggesting that greater individual reduction in right central alpha during sleep restriction
corresponded to greater increase in reported subjective sense of sleepiness in individuals.
Similarly, delta change and KSS were positively correlated, $r(20) = 0.52$, $p = 0.015$,
suggesting that a greater increase in individual delta corresponded with a greater increase
of reported subjective sleepiness in individuals.

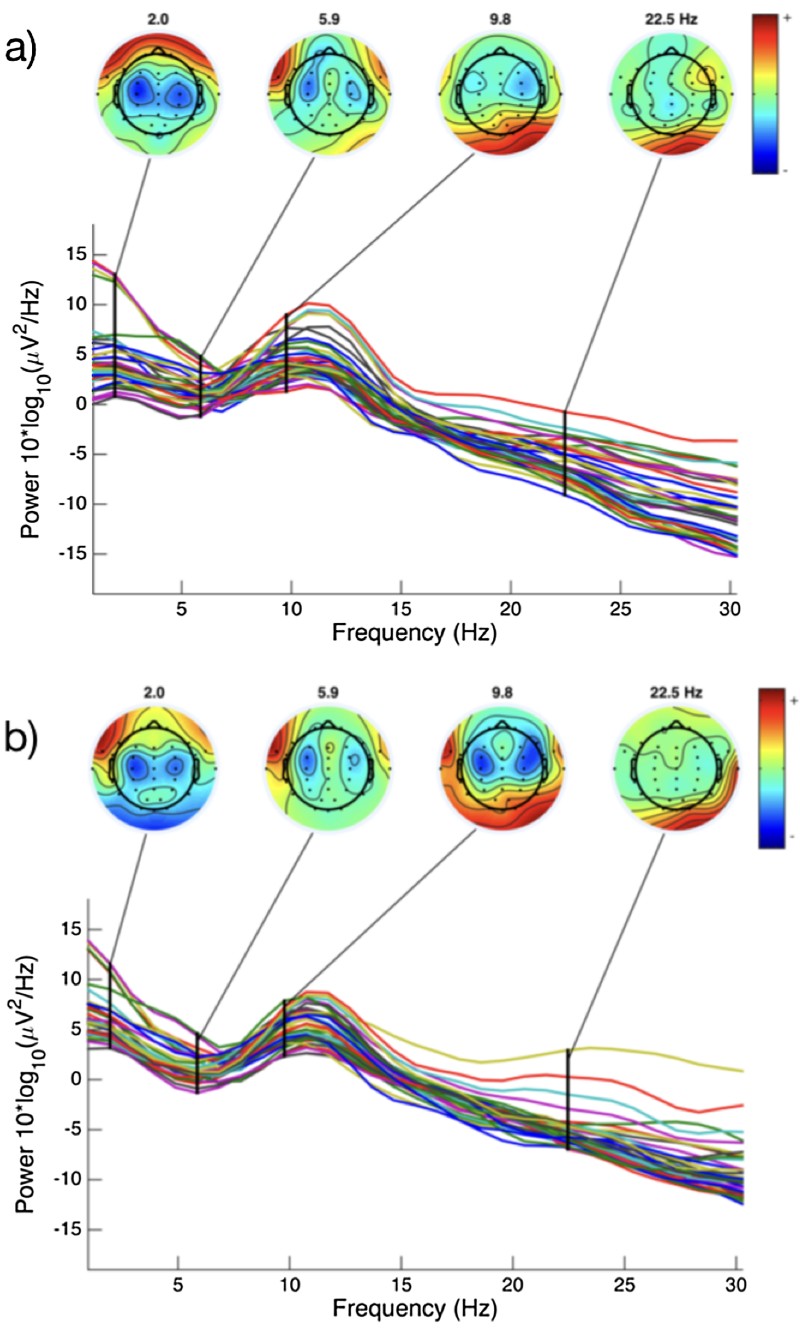

**Figure 7 EEG spectral power for a representative subject.** Variation in spectral power for a single individual during normal sleep (A) and sleep restriction (B) conditions. Figures constructed in EEGLab (*Delorme & Makeig, 2004*).

# GENERAL DISCUSSION

Studies 1 and 2 detected consistent differences between the PVT and AB tasks. Specifically, performance on the PVT was reduced during the sleep restriction condition, whereas performance on the AB task was unimpaired. In addition, the hypothesis that neural oscillations at rest would be related to sleep condition was supported. Alpha oscillations

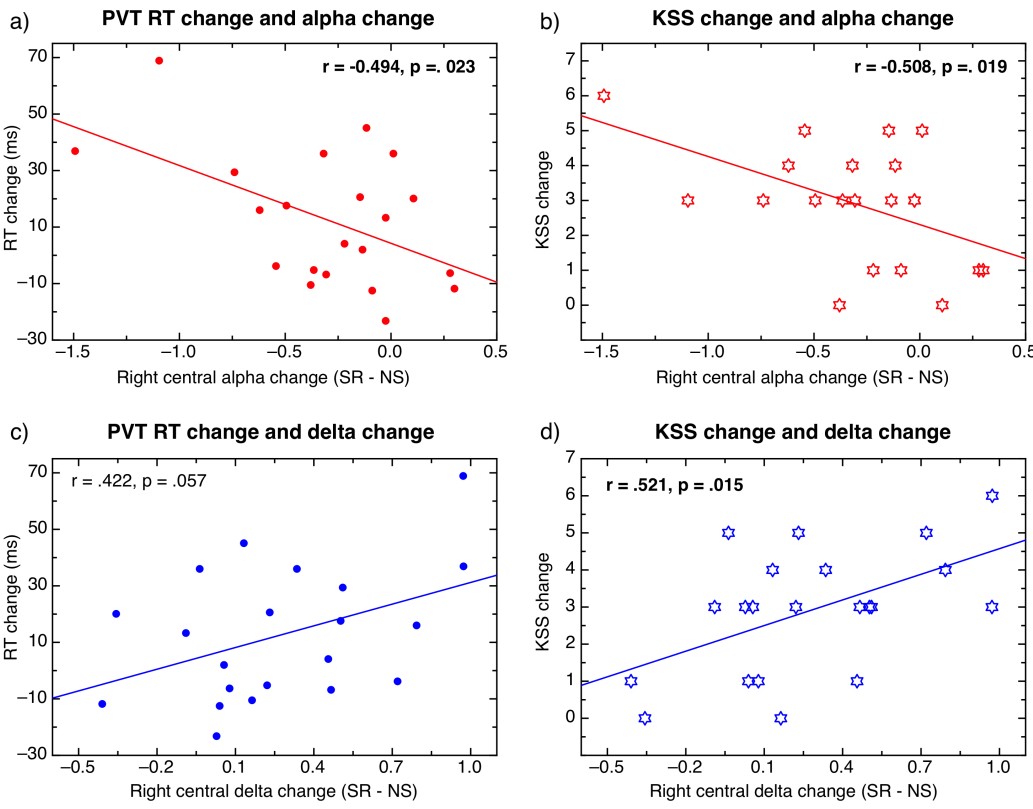

**Figure 8 Correlations between behavioural measures and EEG data.** Correlations between changes in PVT performance (A, C), changes in sleepiness (B, D), and EEG changes in alpha (A, B) and delta (C, D) between normal sleep and sleep restriction conditions.

were significantly reduced during the sleep restriction condition; in particular, centrally in the right hemisphere. The results support the findings of *Strijkstra et al. (2003)* who found that alpha reduced as sleepiness increased. Overall, delta oscillations were not significantly different between sleep conditions, although a marginally significant increase in delta oscillation was observed in the central right hemisphere. As this was the same area where a reduction in alpha oscillations occurred, it is possible that slow wave delta activity superseded the alpha oscillations in the central right hemisphere during sleep restriction. The absence of a significant difference in either theta or delta oscillations might be attributable to the study participants experiencing sleep restriction rather than total sleep deprivation, especially as changes in delta have been consistently found in previous literature (e.g. *Cajochen, Foy & Dijk, 1999*; *Hoedlmoser et al., 2011*; *Strijkstra et al., 2003*).

Two interpretations may account for the preservation of performance accuracy on the AB task during the sleep restriction condition. First, transient attention may be more robust to the effects of sleep reduction than sustained attention. Research conducted by *Shenfield, Beanland & Apthorp (2020)* suggested that the PVT and AB tasks have similar levels of engagement and difficulty, which suggests that sleep loss is unlikely to be

interacting with different levels of difficulty between the two tasks. Thus, it is proposed that preserved AB performance relative to PVT performance during sleep loss is more likely attributable to differences in the cognitive requirements of the discrete forms of attention required for each task (*Braver, Reynolds & Donaldson, 2003*; *Carrasco, 2011*). If this is the case, the studies highlight the need for future research to better understand the impact of sleep loss on discrete forms of attention.

The second potential explanation relates to the reduction in alpha after sleep restriction, which may have prevented a performance decrement on the AB task. This interpretation is supported by previous research which found that lower levels of alpha were associated with better AB performance (*Maclean & Arnell, 2011*; *MacLean, Arnell & Cote, 2012*). It has been suggested that alpha amplitude may interact with the RSVP task specifically, as the stimuli are typically displayed at a rate of 100 ms, which is analogous to the 10 Hz oscillations characteristic of alpha waves (*Mathewson et al., 2011*). Based on this assertion, *Zauner et al. (2012)* proposed that alpha entrainment occurs during the AB, whereby the amplitude of alpha oscillations either facilitates or prohibits the detection of the second target; and they found that alpha amplitude was larger on the trials when an AB was present (i.e. when T1 but not T2 was perceived). Conversely, when alpha amplitude was smaller, they detected a considerable reduction in AB. The analysis of EEG activity during the AB task, as opposed to during a resting state, would be required to provide further support for this theory.

While it is not currently possible to confirm or disconfirm either potential explanation, both of them equally challenge traditional assertions about sleep loss, which propose that attention is a single construct that is reliably impaired during sleep restriction. The findings presented here expand on current knowledge regarding sleep, which has previously focused on measuring neural activity during sleep deprivation (*Borbély et al., 1981*; *Kaida et al., 2006*), as opposed to sleep restriction. This distinction is important, as total sleep deprivation occurs irregularly, and is arguably more representative of a crisis-type situation, whereas sleep loss is relatively common, either consistent or intermittent, due to a person's lifestyle, social and occupational obligations, or physical and mental health conditions (*Banks & Dinges, 2007*; *Knutson et al., 2010*; *Tucker, Dinges & Van Dongen, 2007*). Additionally, this finding has important clinical implications for the non-pharmacological treatment of patients who experience disordered sleep. For example traditional therapeutic interventions addressing sleep (e.g. sleep hygiene, cognitive behavioural therapy, etc.) require clients to employ sustained attention, which has reliably been found to be compromised during sleep loss (*Lim & Dinges, 2008*; *Stepanski & Wyatt, 2003*). The current findings show that sleep loss interventions may be most effective when brief and focused, to better cater to preserved transient attention capacity. These findings also reinforce the importance of safety precautions for tasks which may require sustained attention during sleep loss (e.g. driving; *Beanland et al., 2016*; *Philip et al., 2005*).

# CONCLUSION

Two studies found that performance on a sustained attention task (the PVT) was impaired during sleep restriction; however, performance on a transient attention task (the AB) was unimpaired. EEG activity indicated that differences in neural oscillations, specifically a reduction in right central alpha and an increase in right central delta after sleep restriction, might be driving the observed difference in impairment between the tasks. Both studies detected compelling evidence for the preservation of a discrete form of attention (i.e. transient attention) during sleep restriction. This finding challenges the notion that attention is impacted as a singular construct under conditions of sleep loss, highlighting the possibility that other forms of attention might be robust to the impact of sleep loss. Ideally, the results of this research should shape the way in which findings regarding sleep and attention are expressed in future research, to more accurately describe and investigate the individual components of attention which are being tested and impacted.

# ACKNOWLEDGEMENTS

The authors thank Mille Darvell, Luke Daly and Adrian Wilson for their assistance in data collection for Study 1. Thank you to Dr Rhonda Brown for assistance in the editing and proofreading stages.

## Funding

This work was supported by the National Health and Medical Research Council of Australia (APP1054726) and a QUT Institute of Health and Biomedical Innovation grant. The funders had no role in study design, data collection and analysis, decision to publish, or preparation of the manuscript.

## Grant Disclosures

The following grant information was disclosed by the authors:
National Health and Medical Research Council of Australia: APP1054726.
QUT Institute of Health and Biomedical Innovation.

## Competing Interests

The authors declare that they have no competing interests.

## Author Contributions

- Lucienne Shenfield conceived and designed the experiments, performed the experiments, analyzed the data, prepared figures and/or tables, authored or reviewed drafts of the paper, and approved the final draft.
- Vanessa Beanland conceived and designed the experiments, performed the experiments, authored or reviewed drafts of the paper, and approved the final draft.
- Ashleigh Filtness conceived and designed the experiments, performed the experiments, authored or reviewed drafts of the paper, and approved the final draft.

- Deborah Apthorp conceived and designed the experiments, analyzed the data, prepared figures and/or tables, authored or reviewed drafts of the paper, and approved the final draft.

## Human Ethics

The following information was supplied relating to ethical approvals (i.e. approving body and any reference numbers):

The Queensland University of Technology Human Ethics Committee (Approval no. 1300000793) and the Australian National University Human Ethics Committee (#2015/184) granted ethical approval to carry out these experiments.

## Data Availability

The data are available on the OSF: Shenfield, Lucienne, Vanessa Beanland, Ashleigh Filtness and Deborah Apthorp. 2020. "Sleep Loss, Attention and EEG." OSF. April 17. osf.io/e57pc.

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
