# Peer review of "The impact of sleep loss on sustained and transient attention: an EEG study"

_PeerJ, doi:10.7717/peerj.8960_

## Round 0.1 · original submission · Major Revisions

We ask to pay particular attention to the comment of R2 on the use of actigraphy as a measuring instrument in the manuscript.

Reviewer 1 ·

Basic reporting

1)Summary: The summary needs to be re-written as it is not clear enough.
- Particulalry, please remove the term "novel finding" as it is now adequate here, given the rest of the summary.
- Also, please be more precise with the EEG method.
- The relationship between teh results described and the conclusion are not clear enough and this should be modified.

2) Introduction
- The paragraph from ilne 46 ("Considerable empirical evidence...") is nto clear. Why are you referring to the central nervous system? I also miss a clear conclusion related to Owen 2009, more precision are needed.
- I miss a reference related to the effects of sleep loss on attention.
- Why are you talking about fatigue? This is different from vigilance/attention etc, please remove as it is not appropriate.
- I miss references and global conclusions related to the effects of sleep loss on both the PVT and the AB.
- EEG paragraph: The first sentence lacks precision. The sentence line 102 ("However, the majority") lacks a conclusion and more precise information related to the importance of sleep restriction versus total SD. I also miss the effects of the one versus the other.
- In the EEG part, I also miss the link between sleep loss and its effects on EEG, notably on alpha.
- You refer to AB performance in the middle of various forms of attention, pelase correct (line 112).

Experimental design

Study 1
2) Methods:
- Did you assess actigraphy? If yes, it should be reported.
- According to me, training shoud be done 1 week before the actual study and not just before, this can create a bias and authors should aknowledge for this is the discussion.
- The PVT description should be more precise.
- In the statistical analyses, did you check for noramality and homoscedasticity and adapted your statistical analyses in function?

3) Results:
- You did not assessed sleep but actigraphy that record sleep-wake alternation. This is different and should be changed in all the manuscript.
Please replace sleep restriction by SR in the PVT part.

4) The discussion is very short and the second paragraphe should have been in the introduction. Please adapt the manuscript according to these comments (ex: maybe remove the discussion for the Study 1).

Study 2:
2) Methods:
- It is not clear if you used the same participants?
- I am not sure about the valiidity of FitBit, do you have references?
- Why did you use a dark room in the second study and not in the first one? This could consitute a bias. Surprisingly, the second study is very precisely described in its method compared to the first one.
- EEG please add the name of the reference and its placement.
- Why did you use PSychophysics for Study 2 and not study 1? What about the timing accuracy in both cases?
- Wy did the PVT changed?
- Why EEG was done eyes closed and not eyes open? What would have been different?
- I miss references related to each method point and its justification in the EEG paragraph. I also miss an explanation related to the lack of analyses on other frequency bands.
- Why did you use a sliding window for analyses? And not a global analyse? The EEG part is really unclear to me, particulalry in the justification of methods used.
- Statistical analyses: Which factors did you use in the ANOVA?

Validity of the findings

3) Results:
- Be aware that you are not assessing sleep and correct for this, please.
- Please remove all part related to methods (normality etc) and place it in the appropriate section.
- Please be clear that EEG was completed before task and eyes closed!

4) Discussion
- First paragraph: if you refer to tiredness, please relate it to KSS results instead of alpha ones.
- The explanation line 493 is not clear and seem wrong to me. Please remove or adapt (slow wave and alpha).
- Please be more specific about the consistent findings in previous studies.
- Please remove the sentence line 507, this is not useful.
- The second "explanation" is not an explanation, this is an interpretation of you results. Please modify.
- The last pragraph is nto useful ad partly inappropriate, please remove.

Additional comments

The current paper report results from 2 studies related to the effects of sleep loss on attention.
The paper is globally clear but lack precision at several points. In addition, some methodological issues need to be discussed.
General comment in all the manuscript: Please be more specific when you can, also more precise and with references at each point. Also, please remove the term "sleep" and be clear that EEG was done before task completion and eyes closed.

·

Basic reporting

The manuscript appear to be clear and unambigouous in the writing. English (as it appears to me) is professional.
Literature references are complete, giving a sufficient scenario of the reach background of this reserach field.
The structure of the article conforms to an acceptable format of ‘standard sections’. Figures are surely relevant, but in my opinion, they should be reduced in number.
Hypotheses are weel tested and "answered".

Experimental design

Aims and scopes of the manuscript fall within the area of interest of the journal.
The manuscript clearly defines the research question, that results to be relevant and meaningful. As a consequence, the knowledge gap is univocally identified.
Methods are clearly described, so that replication is possible.

Nonetheless this, I have serious concerns related to the choice to use FitBit as a reliable measure of actigraphy. Actigraphy is a well-known and accepted measure in psychophysiology; it can provide a direct measure of motion and activity and a sufficiently reliable indirect measure of sleep.at the same time it is surely useful as a compliance index when deprivation studies arre carried out. FitBit in my opinion is absolutely not acceptable as a professional measure of sleep, and the authors did in study 2 (section 7.1).
As a consequence, I strongly suggest the AUthors, to delete all the section in which a commercial device is proposed as a professional technique.

Validity of the findings

Good level of impact and noveltu/originality of the findings.
Results are statistically sound and controlled so that findings can be generalized.
Conclusions well stated and linked to the original question.

Additional comments

In general it is my opinion that the present manuscript is of interest for the readers of the journal and that can be easily imporved and made acceptable for publication.

---

## Round 0.2 · Minor Revisions

Please include the very minor suggestion from Reviewer 1

Reviewer 1 ·

Basic reporting

All comments have been adressed.

Experimental design

All comments have been adressed.

Validity of the findings

All comments have been adressed.

Additional comments

I would like to thank authors as they adressed all my comments. Regarding the sleep term, I feel that the reader will be aware of the limit of using actigraphy as an objective measure of sleep (maybe authors could just add a small sentence related to this) and I accept the modifications that have been done.

·

Basic reporting

The Authors managed all the concerns I raised in my previous reviewing. As a consequence tha manuscript can be published in its present form.

Experimental design

see point 1

Validity of the findings

see point 1

Additional comments

see point 1

---

## Round 0.3 · accepted · Accept

Thank you for following the suggestions and replying to all comments received. The manuscript has improved, in my opinion.